# The Roles of Mepiquate Chloride and Melatonin in the Morpho-Physiological Activity of Cotton under Abiotic Stress

**DOI:** 10.3390/ijms25010235

**Published:** 2023-12-23

**Authors:** Yanqing Wu, Jiao Liu, Hao Wu, Yiming Zhu, Irshad Ahmad, Guisheng Zhou

**Affiliations:** 1Joint International Research Laboratory of Agriculture and Agri-Product Safety, The Ministry of Education of China, Institutes of Agricultural Science and Technology Development, Yangzhou University, Yangzhou 225009, China; yqwu@yzu.edu.cn (Y.W.); jiaoliu0407@163.com (J.L.); w13964483386@163.com (H.W.); 18136059863@163.com (Y.Z.); 2Jiangsu Provincial Key Laboratory of Crop Genetics and Physiology, Yangzhou University, Yangzhou 225009, China

**Keywords:** mepiquate chloride, melatonin, abiotic stress, morpho-physiological activity, yield, cotton genes

## Abstract

Cotton growth and yield are severely affected by abiotic stress worldwide. Mepiquate chloride (MC) and melatonin (MT) enhance crop growth and yield by reducing the negative effects of abiotic stress on various crops. Numerous studies have shown the pivotal role of MC and MT in regulating agricultural growth and yield. Nevertheless, an in-depth review of the prominent performance of these two hormones in controlling plant morpho-physiological activity and yield in cotton under abiotic stress still needs to be documented. This review highlights the effects of MC and MT on cotton morpho-physiological and biochemical activities; their biosynthetic, signaling, and transduction pathways; and yield under abiotic stress. Furthermore, we also describe some genes whose expressions are affected by these hormones when cotton plants are exposed to abiotic stress. The present review demonstrates that MC and MT alleviate the negative effects of abiotic stress in cotton and increase yield by improving its morpho-physiological and biochemical activities, such as cell enlargement; net photosynthesis activity; cytokinin contents; and the expression of antioxidant enzymes such as catalase, peroxidase, and superoxide dismutase. MT delays the expression of *NCED1* and *NCED2* genes involved in leaf senescence by decreasing the expression of ABA-biosynthesis genes and increasing the expression of the *GhYUC5*, *GhGA3ox2*, and *GhIPT2* genes involved in indole-3-acetic acid, gibberellin, and cytokinin biosynthesis. Likewise, MC promotes lateral root formation by activating *GA20x* genes involved in gibberellin catabolism. Overall, MC and MT improve cotton’s physiological activity and antioxidant capacity and, as a result, improve the ability of the plant to resist abiotic stress. The main purpose of this review is to present an in-depth analysis of the performance of MC and MT under abiotic stress, which might help to better understand how these two hormones regulate cotton growth and productivity.

## 1. Introduction

Cotton is a cash crop cultivated for the textile industry, accounting for 35% of global fiber consumption [1]. Major cotton-producing countries include the United States, China, India, Pakistan, and Brazil [2]. Cotton’s growth and productivity are severely impacted by abiotic stress, which decreases lint yield and fiber quality [3,4]. Abiotic stress, caused by a variety of adverse environmental conditions, such as cold, salt, heavy metals, drought, and high temperature, leads to a series of morpho-physiological, biochemical, and molecular changes in plants that adversely affect plant growth and yield [1]. China is the world’s largest cotton producer; however, some parts of the country, like Xinjiang, are frequently impacted by cold stress, which persists for more than half of the cotton growth season [5]. Current research and long-term production practices worldwide, particularly in Xinjiang, reveal that cold damage significantly impacts cotton growth and yield during the seedling stage [6]. Therefore, it is essential to study cotton morpho-physiological, biochemical, and molecular responses during seedling growth under low temperatures and to introduce new methods for promoting abiotic tolerance in cotton [7].

Different crops, such as cotton (*Gossypium hirsutum* L.), maize (*Zea mays* L.), soybean (*Glycine max* L.), rice (*Oryza sativa* L.), and tomato (*Solanum lycopersicum* L.), cannot adapt to their environment when confronted with abiotic stress [8,9,10]. Plants have evolved complex mechanisms by incorporating plant transcription factors to cope with the adverse effects of abiotic stress [11,12]. Plants have coping mechanisms that integrate biochemical, physiological, and molecular systems and stress detection, endogenous hormones, signal transduction pathways, and gene identification to protect or mitigate the adverse effects of cellular oxidative damage [13].

The study of cotton abiotic stress tolerance regulation has significantly increased in recent years. The hormones MT and MC have been recently found to be among cotton’s most efficient abiotic stress signaling regulators [1,14]. As a result of suppressing gibberellin (GA) production in plants, it has been shown that MC improves cotton yield by reducing leaf area and plant height and reducing the length of shoots and fruit branches [15,16,17]. Applying MC affects the physiological characteristics of cotton leaves [18]. Cotton plants treated with MC had higher leaf weights, thicker leaves, chlorophyll content, and net photosynthetic activity [19]. In cotton, MC increased the number of photoassimilates produced by the leaves, altered the source–sink relationship, decreased ribulose 1,5-bisphosphate carboxylase/oxygenase (RuBisCO) activity, and increased stomatal conductivity [20]. MC application has been shown to increase the functional stage of cotton leaves by expanding the content of active cytokinin (CK) in the middle and final phases of leaf growth, decreasing GA synthesis, delaying the peak time of abscisic acid (ABA) and ethylene (ET), and delaying leaf senescence [21]. However, the role of MC in these morpho-physiological activities of cotton during abiotic stress remains unknown.

Most animals and plants also produce MT in their mitochondria and chloroplasts [22]. As a result, cotton seeds are predicted to contain endogenous MT [23]. Previous research has shown that MT possesses excellent antioxidant properties and can improve crop resistance through signal modulation under abiotic stress [24]. MT is engaged in various processes in plants, including root and flower growth, leaf senescence, fruit ripening, photosynthetic activity, and stress-induced oxidative damage elimination [25,26,27]. MT enhances the expression of superoxide dismutase (SOD), catalase (CAT), glutathione peroxidase (GSH-PX), glutathione reductase (GSSG-R), and ascorbate peroxidase (APX), which scavenge free radicals and provide cold resistance in cotton [28]. MT metabolites also scavenge the overproduction of reactive oxygen species (ROS) and reactive nitrogen species (RNS), distinguishing them from most other antioxidant enzymes [29]. Under cold stress, MT enhances antioxidant activity by enhancing gene expression and redox status [9,30]. Moreover, MT enhances cold resistance by increasing electron transport and antioxidant activity and by degrading starch [31]. Several MT functions have been demonstrated, many of which contribute to stress tolerance in different crops [32]. For example, MT application substantially reduced electrolyte leakage in tomatoes under cadmium stress (Cd) but had no effect under normal conditions [33]. The appropriate rate of MT not only accumulated and promoted the total protein content in *Malus hupehensis* but it also accumulated the total protein content in kiwifruit leaves while boosting cell fluid concentration and stress resistance [34,35]. Exogenous MT promoted soybean growth and increased protein content under salt stress [36].

Different studies have shown that MC- and MT-treated plants and their relevant genes could alleviate the detrimental effects of abiotic stress, thus indicating that these hormones are beneficial for agricultural yield (Table 1). Nevertheless, the performance of MC and MT in enhancing these mechanisms under abiotic stress, especially in cotton crops, is still far from clear. Therefore, the primary purpose of this investigation is to provide an in-depth analysis of the impact of MC and MT on morpho-physiological and biochemical activities of plants and their relevant genes under abiotic stress in cotton, as well as their respective functions, signaling, and transduction pathways.

## 2. Biosynthesis, Signaling, and Transduction Pathways of MC and MT

### 2.1. Synthetic Pathway of MC

MC is a compound used to regulate plant growth and development because it is absorbed by the plant’s leaves and then distributed to the other parts of the plant [51]. MC is a water-soluble exogenous hormone that may be supplied by soaking seeds or spraying leaves [52]. Previous research demonstrated that MC controls plant growth by inhibiting GA biosynthesis [53]. GA is a hormone that promotes the growth of internodes and stems by cell division and expansion [54]. MC blocks the ent-copy diphosphate synthase (CPS) and ent-kaurene synthase (KS) during the initial stages of GA metabolism [52]. Moreover, the impact of MC on GA biosynthetic genes is caused by either interacting with the GA biosynthesis pathway or by upregulating the GA repressor, DELLA [55]. MC treatment resulted in the downregulation of cell-loosening genes, like *GHEXP* and *GhXTH2*, which led to a considerable decrease in the endogenous level of bioactive GAs like GA_3_ and GA_4_ [16]. The reduction in the bioactive content of GAs is related to the MC-induced downregulation of GA biosynthetic genes, including *GhGA3ox*, *GhGA20ox*, *GhCPS*, and *GhCPS* [56]. A previous study on cotton seedlings found that MC reduced the gene expression involved in GA metabolism and signaling pathways, decreased the contents of GA, and hindered cell elongation [57]. Nevertheless, the entire transcriptional regulatory mechanism underlying MC-mediated growth inhibition is still unclear [16].

MC regulates a wide range of biological activities in various crops [58]. In response to abiotic stresses, MC decreased vegetative growth while increasing crop yield in soybean and cotton under drought stress [40,59]. Exogenous MC treatment in cotton can enhance fiber growth and production while reducing the leaf area, stem elongation, and plant height [60]. Previous research has shown that compared to other crops, including maize, wheat, and soybean, cotton is more susceptible to MC for unknown reasons [56,61]. Hence, MC is often used in cotton to control cotton development, such as by reducing the internode length and leaf area while increasing the fiber quality and yield [19,62,63]. MC treatment has become one of the most essential agronomic strategies in commercial cotton production due to its beneficial effect on plant growth and yield. In this regard, more research is needed to better understand the effects of MC on cotton growth and fiber yield, its association with GAs, and its essential genes in cold-stressed cotton plants.

### 2.2. The Signaling and Transduction Process of MC

MC increases the growth and yield of cotton via signal transduction [52]. MC application slowing GA activity involved in cell elongation restricts signaling and transduction pathways activation, hinders vegetative development, and alters GA homeostasis by activating site-specific genes, reducing plant height ([55], Figure 1). MC inhibited soybean growth by decreasing the concentrations of GA, brassinolide, zeatin, and other plant hormones and signaling pathway-related genes [40]. MC affects the signaling of various hormones [16]. This variation has a more significant effect on sensitive crop varieties. For example, a previous study demonstrated that numerous tryptophan metabolism genes and all genes in the auxin-responsive *GH3* gene family are downregulated in the HN65 drought-resistant soybean variety [64]. The *GH3* gene family controls plant and cell growth in different crops. During the inhibition of cell division and zeatin production, the expressions of the first few genes were downregulated and the genes responsible for GA and brassinosteroid biosynthesis were suppressed. The downregulation of GA and brassinosteriod gene expression may significantly decrease crop growth and yield [64].

Due to the acceleration of IAA conjugate hydrolysis and the consistent upregulation of *GhLBD18-1*, *GhLBD18-2*, *GhARF19*, *GHLBD18-1*, and *GhARF7* gene expression in the roots of cotton seedlings, the MC treatment of cotton seeds could increase the content of IAA and, as a result, increase the number of lateral roots (LRs) [65]. IAA is pivotal in controlling pericycle cell priming and growth [66]. It may be produced in cell biofilms, enhancing the plant’s antioxidant defense system and scavenging the ROS free radicals in plants during abiotic stress [67].

Numerous studies have shown that MC increases stress resistance by enhancing the physiological and metabolic processes of plants, such as protein, sugar, and chlorophyll activity, accumulating free amino acids, SOD, CAT, and POD, and decreasing MDA activity in sunflower leaves [68]. Increases in the concentrations of enzymes such as CAT, POD, and SOD, osmotic adjustment (soluble sugar and proline) in the leaves, and ABA accumulation (which mitigates the harmful effects of cold stress) are all evidence that exogenous MC improves the growth and cold tolerance of sweet pepper and cotton [38,55]. Stress resistance increased significantly due to an increase in ABA content. However, the performance of MC in these mechanisms in cotton under abiotic stress is still unknown.

### 2.3. The Biosynthetic Pathway of MT 

Plants, bacteria, and mammals all produce MT, a biogenic amine. The “phytomelatonin” was first introduced by [69] to refer to the substance’s plant origins after it was first detected in higher plants in the 1990s. MT was identified and quantified in over 20 monocotyledonous and dicotyledonous crops [70,71]. According to numerous research investigations, tryptophan has been identified as the primary substrate involved in the first stages of MT production. Tryptamine 5-hydroxylase, tryptophan hydroxylase, N-acetylserotonin methyltransferase, serotonin N-acetyltransferase, tryptophan decarboxylase, and caffeic acid 0-methyl transferase are six enzymes involved in the four-step enzymatic process [49,72]. In the first two steps, tryptophan decarboxylase catalyzes the breakdown of tryptophan to generate tryptamine.

Furthermore, tryptamine-5-hydroxylase (T5H) catalyzes the conversion of tryptamine to serotine; serotine is the most critical step in MT synthesis [73]. Serotonin N-acetyltransferase catalyzes the conversion of serotine to N-acetylserotonin, which N-acetylserotonin methyltransferase then converts into MT. Zhang et al. [46] found that exogenous MT controlled 52 SNAT genes in the *Gossypium hirsutum* genome and a subset of *GhSNATs* under salt stress. The *GhSNAT3D* gene-silenced plants had lower endogenous amounts of MT, Ca^2+^, and antioxidant enzymes and decreased salt tolerance. Exogenous MT-silenced plants had increased endogenous levels of MT, Ca^2+^, and antioxidant enzymes and improved the salt resistance of *GhSNAT3D* genes. *GhSNAT3D* genes may further interact with *GhSNAT25D* and N-acetylserotonin methyl-transferase to regulate MT synthesis in cotton under salt stress [46]. However, the underlying molecular mechanism in cotton under abiotic stress remains unclear.

Additionally, HsFA1a overexpression in tomatoes boosted endogenous MT accumulation and the expression of the MT biosynthesis gene caffeic acid O-methyltransferase1, which resulted in an increase in Cd stress [74]. MT is essential for growth control, serotonin production, and abiotic stress tolerance [75]. More research is required on the performance of MT in serotonin in plants, especially in cotton, even though the existence and function of MT in serotonin is a prominent developing research topic. The first stage in the MT biosynthesis process in transgenic rice is the decarboxylation of tryptophan catalyzed by tryptophan decarboxylase [76]. The *T5H* genes are required for serotonin production. The expression of *T5H* genes increases the MT amount in transgenic rice, indicating that the MT amount in plants is not dependent on serotonin [72]. Exogenous MT increases the amount of endogenous MT in cotton under salt stress and decreases the *T5H* gene expression responsible for serotonin production in rice. However, the increment in the MT amount and the upregulation or downregulation of *T5H* in cotton under other abiotic stresses are still unknown.

### 2.4. The Signaling and Transduction Pathway of MT

MT is a signaling compound that mitigates the negative effects of abiotic stress by activating the CK, ET, salicylic acid (SA), ABA, and GA signaling pathways to induce several defense genes ([22], Figure 2). GA and ABA are pivotal plant hormones that play a vital role during abiotic stress. Under salt stress, MT-treated *Limonium bicolor* seeds significantly increase seed germination because of increased endogenous MT, GA, and ABA contents [30]. MT increases these hormones, particularly ABA production, under salt stress, activating several signaling pathways and upregulating transporter genes, such as *HKT1*, *AKT1*, and *NHX1* [77]. MT treatment improves the expression of SOS1 in NaCl-treated roots, which may help to facilitate sodium (Na^+^) export from the roots and store it in stems, preventing Na^+^ from reaching the photosynthetic leaf tissue [78].

MT also enhances plant antioxidant capacities that scavenge the production of ROS and RNS [50]. MT at the plasma membrane causes cytosolic Ca^2+^ to increase and signals hydrogen peroxide (H_2_O_2_) accumulation through nicotinamide adenine dinucleotide phosphate (NADPH) oxidase to scavenge ROS and RNS [36]. In tomato fruit, signaling H_2_O_2_ accumulation treated with MT for seven days under cold stress may serve as a defense response to cold stress by promoting the endogenous accumulation of MT. Furthermore, reducing harmful H_2_O_2_ accumulation after seven days of cold stress in tomato fruits treated with MT alleviates oxidative stress, possibly resulting from endogenous MT accumulation with enhanced ROS scavenging properties [79]. In contrast to tocopherols, glutathione antioxidants, and ascorbic acid, which can only scavenge one ROS per molecule, MT shows a ROS scavenging cascade, allowing one MT molecule to scavenge up to 10 ROS [79]. This confirms that MT signaling improves the crop morpho-physiological and antioxidant activity under abiotic stress by different plant processing. However, the performance of MT signaling via this process needs to be further investigated in cotton under abiotic stress.

## 3. Role of MC and MT in Plant Growth, Biochemistry, and Yield under Abiotic Stress in Cotton

### 3.1. Role of MC in Cotton Growth, Physiology, and Yield 

MC is a plant growth regulator that affects plant structure and function by using multiple biochemical functions and modes of action to influence crop growth and development [55,80]. MC decreased the internode length, number of nodes, canopy, leaf area, and plant height but increased light interception and, as a result, increased yields ([60], Figure 3). The reduction in morpho-physiological activity is triggered by a decrease in the GA concentration caused by MC application in plants; as a consequence, MC disrupts cellular movements due to decreased cell wall relaxation and increases cell wall stiffness and plasticity [55]. GA content inhibits cell elongation, which restricts the stem’s vertical growth [81]. MC impacts cotton plants in ways such as decreasing the stem and leaf growth but increasing the maturity [21]. The MC treatment balances vegetative and reproductive development in cotton, improving the fiber quality and yield [82].

MC treatment affects the physiological activity of cotton leaves [83]. Cotton plants had enhanced net photosynthetic activity by producing thicker weight leaves with higher chlorophyll activity [21]. The most crucial metabolic activity influencing plant growth and yield is photosynthesis. The reduction in photosynthetic activity under MC application depends on stomatal and non-stomatal restriction. However, non-stomatal factors have a higher role in restricting photosynthetic activity [84]. During net photosynthesis, Rubisco plays an indispensable performance in regulating the Calvin cycle and fixing carbon dioxide (CO_2_) [85]. The exogenous MC affects the activity of Rubisco enzymes. Reddy et al. [19] demonstrated that MC treatment reduced the activity of Rubisco, which was strongly associated with decreased photosynthetic activity. Other studies have shown that MC application suppressed the activity of Rubisco enzymes because of the drop in CO_2_ fixation, which caused a decrease in the net photosynthetic activity [86,87]. According to different research, MC application inhibited the activity of the Rubisco enzyme and hindered photosynthetic activity. However, the current investigation shows that the effects of MC on the Rubisco enzyme and its relationship with the net photosynthetic activity in cotton under abiotic stress are still unknown.

The timing and dose of MC directly influence the cotton fiber quality and yield [60]. Depending on the application timing, genotypes, MC application dose, environmental conditions, and other management practices, MC may have decreased [88,89], increased [90], or not affected the cotton yield [91,92].

Different research has been introduced to identify the stress tolerance mechanism in cotton. Different research has determined whether MC alleviates the detrimental effects of various abiotic stresses. For instance, according to [37], cotton growth and productivity increased using MC as a seed pretreatment agent when exposed to salt stress. Similarly, MC application increased cotton’s solvent protein, free proline, and chlorophyll pigments but reduced its MDA contents and, as a result, improved its pressure resistance [55]. MC treatment dramatically increased sweet pepper growth by enhancing ABA accumulation and increasing the amount of osmotic-regulating compounds, including soluble sugar and proline, under cold stress [38]. MC reduced the adverse effect of drought stress by improving the expression levels of flavonoid differential metabolite in soybeans [64]. Different studies have shown that MC reduces the adverse effects in different crops via improving plant morpho-physiological activity under abiotic stress. However, the significance of MC in the morpho-physiological activity and yield of cotton under cold stress remains unknown. Furthermore, it is unknown how MC regulates the growth and productivity of cotton under cold stress at the molecular level.

### 3.2. Role of MT in Cotton Growth, Physiology, and Yield under Abiotic Stress

Plants are frequently subjected to abiotic stresses [49]. For example, cold stress inhibits average plant growth and, as a result, decreases crop yields ([13], Figure 4). There are numerous methods for decreasing the detrimental effects of cold stress on corps. Exogenous MT is one of the most prominent techniques that reduce the adverse impact of cold stress in various crops [50]. MT has made significant contributions to the current agriculture systems. However, MT has generated some adverse effects on our natural environment. Therefore, it is crucially important to thoroughly explain the specific mechanism of exogenous hormones in cotton to prevent the natural environment from increasing cold stress harm.

Cold stress severely reduces crop yield by causing significant decreases in plant physiological activity, such as low photosynthetic efficiency and an imbalance of mineral nutrition [93]. Cold stress significantly increases lipid peroxidation, electrolyte leakage, and malondialdehyde (MDA) [94]. However, the treatment of MT significantly decreases MDA formation and ROS deterioration on the cellular membrane of cold-stressed rice seedlings and, as a result, improves plant growth and productivity [95]. MT is an essential compound that improves crop development and yields by alleviating the harmful effects of abiotic stress [96,97]. Different studies have demonstrated that exogenous MT could reduce the detrimental effects of salt stress in watermelons, drought stress in *Moringa oliguria*, and cold stress in cucumbers [78,98,99]. New research on cotton seedlings showed that exogenous MT controlled leaf photosynthesis activity, leading to considerably greater fresh and dry weights than control under Cd stress [42]. The improvement in the growth and functioning of the leaf was strongly associated with the MT-induced regulation of Cd transporter genes in cotton roots, such as *LOC107894197, LOC107955631,* and *LOC107899273*. This indicates that MT causes the Cd transporter genes to downregulate to prevent Cd ion transfer into the leaf tissue [42]. Similarly, Xu et al. [100] showed that treating radish plants with MT mitigated Cd-induced toxicity via modulating Cd transporter genes and heavy metal stress to scavenge harmful ROS.

MT regulating plant growth and yield in barley, oat, and wheat showed that MT increased coleoptiles by 10 to 55% [101]. Additional studies on MT in transgenic rice plants revealed that it enhances seedling biomass, root development, and root biomass at early stages, contributing to higher yields [102]. Applying MT might improve growth traits and yield by promoting cell expansion, decreasing osmotic stress, and retaining water contents in the shoots [103]. Likewise, [104] showed that exogenous MT could remove the negative effects of cold stress and enhance plant growth and productivity. Different research has demonstrated that MT treatment improves growth characteristics and yield in plants due to enlargement in the cell, decreases the effect of osmotic stress, maintains the relative water contents in shoots under cold stress, and regulates plant photosynthetic activity under Cd stress due to the downregulation of Cd transporter genes by inducing more Cd ion transporters towards the cotton leaf tissue. However, the process by which MT improves these outcomes in cotton under abiotic stress is still poorly understood.

### 3.3. Role of MC in Plant Defense Systems 

Soluble protein and sugar are critical components of plant defense mechanisms under cold stress [105]. Due to their higher water absorption, soluble protein and sugar are essential osmotic regulatory substances that may control crop growth and productivity by lowering the chilling stress level in cell liquids and reducing the water loss capacity of the protoplasm [106]. MC enhances soluble protein and sugar content in cotton leaves. Nevertheless, the performance of MC in increasing these osmotic regulatory chemicals under cold stress in cotton still needs to be clarified.

Plants under cold stress can trigger the removal of excessive excitation energy via self-regulation [107]. However, due to the limitations of self-regulation, there is still a percentage of excitation energy that cannot be efficiently used or dispersed. Some of this energy converts to electric energy, which causes an excessive amount of ROS to be produced [108]. Higher ROS react with the plasma membrane to create MDA, which affects different physiological processes, such as the Calvin cycle, and damages plant membranes, DNA, protein degradation, enzyme passivation, and chloroplasts or causes other organ deformation [109]. MC may successfully reduce the production of ROS caused by different stresses in cotton, including salt, Cd, and drought stress [37,109]. Previous research has shown that MC increases plant defense by increasing the antioxidant enzyme activity and lowering MDA formation in cold-stressed tomato plants [109]. However, further investigation into the biochemical and molecular processes of MC in crop production is necessary to alleviate abiotic stress in cotton.

### 3.4. Role of MT in Plant Defense Systems 

As an internal sensor of oxidative damage, MT is considered the first line of defense in crops [110]. Endogenous MT not only neutralizes RNS and reactive ROS but it also activates plant antioxidant enzymes and increases the antioxidant capacity of the plant [111]. Exogenous MT increases the activity of plant antioxidant enzymes, such as SOD, CAT, POD, and APX, which improves the capacity of plant antioxidant enzymes and mitigates the adverse effects of cold stress in cotton, cucumber seedlings, and tea trees [31,112]. Previous research has demonstrated that exogenous MT increased plant photosynthetic C assimilation by improving the antioxidant activity of organelles under cold and drought stress in cotton and barley [44,112]. MT may improve plant life by increasing starch metabolism and energy supply in response to ROS damage during cold stress [113].

MT regulates higher ascorbic acid (AsA) and GSH content and reduces oxidized glutathione (GSSG) and dehydroascorbic acid (DHA) content in cotton crops, which delays leaf senescence under dark conditions [23]. GSH and AsA are key reducing chemicals that protect plants against membrane lipid peroxidation. The AsA-GSH cycle system has successfully scavenged free radicals in different crops [114]. Under abiotic stress, it is crucial to increase the antioxidant defense system in plants through MT application [115]. MT has been shown to increase plant development and growth by increasing its antioxidant defenses against abiotic stress. Nevertheless, the pivotal performance of MT in improving the antioxidant defense system in cotton crops under abiotic stress is still unclear.

## 4. MC and MT Relevant Gene Involved in Cotton Growth

### 4.1. MC Relevant Genes Involved in Cotton Growth

MC application inhibited cell elongation in cotton plants by downregulating the *GhEXP* and *GhTH2* genes due to a decrease in the endogenous GA_3_ and GA_4_ levels in the elongation internode, resulting in a reduction in plant height ([116], Figure 5). Within 2 to 10 days after MC treatment, the GA metabolic and biosynthetic genes were dramatically suppressed, downregulating the expression of DELLA-like genes [116]. GAs affect cell division and elongation to control the stem and internode lengths. GAs increase plant growth by promoting DELLA protein degradation under cold stress, which are important repressors of GA signaling, by interacting with the gibberellin insensitive dwarf1 (G1D1) receptor via the ubiquitin proteasome pathway [117,118]. When the GA concentration is reduced, DELLA proteins interact with transcription factors (TFs) to inhibit their target gene expressions and promote cell division and expansion under abiotic stresses, such as drought, salt, and high-temperature stress [16,119,120].

MC induces lateral root formation, which enhances root development. MC facilitates the growth of lateral roots, possibly by controlling the endogenous GA [52]. The results of an RNA-seq study showed that MC upregulates the expression of the GA catabolism gene *GA20x* while downregulating the expressions of *GA30x*, *GA2oox*, and *CPS*, which are involved in GA biosynthesis [52]. Consistently, the concentration of GA in the root was significantly lower in the MC-treated seeds as compared to the control. The GA receptor, GA1D1, causes DELLA protein degradation and further controls the *XERICO* gene’s production, a stimulator of ABA biosynthesis [121]. This suggests that MC application during the seed’s developmental stage might decrease the endogenous levels of ABA due to DELLA protein degradation under drought stress, which controls the production of the *XERICO* gene [122]. Similar results were demonstrated by [52], who reported that MC application significantly increased the auxin concentration, reduced the ABA content in the root tissue, and increased later root growth under abiotic stress [123]. It was demonstrated that higher ABA concentrations suppressed the upregulation of auxin transporter genes, such as *PIN1*, *PIN3*, *PIN5*, *PIN7*, and *AUX1*, in the roots by decreasing auxin concentrations, resulting in root growth inhibition [124]. Nevertheless, the underlying molecular process of this phenomenon in cotton under other abiotic stresses is still unknown.

### 4.2. MT Relevant Genes Involve in Cotton Growth

Cotton’s photosynthetic activity decreases as a result of leaf senescence, which has a significant influence on fiber growth and production under salt stress ([125], Figure 6). Previous research has shown that exogenous MT helps to prevent leaf senescence under drought stress conditions [112]. The chlorophyll degradation-related genes, such as *GhNAC12* and *GhWRKY27/71*, were dramatically downregulated by MT application [126]. Interestingly, by interacting with other plant hormones, MT plays an indispensable role in maturing under abiotic stress, including cold stress in different crops [127]. A previous study demonstrated that ABA is the primary hormone that promotes leaf senescence under abiotic stress in different crops [128]. The content of ABA and its biosynthesis genes gradually increased during leaf senescence in cotton plants [58]. MT reduced the upregulation of ABA biosynthesis genes, *NCED1* and *NCED2*, and their accumulation in cotton [129]. Similarly, exogenous MT delayed leaf senescence via the upregulating genes involved in IAA, GA3, and CTK biosynthesis, such as *GHIPT2*, *GhYUC5*, and *GhGA3ox2*, thereby increasing the endogenous levels of IAA, GA3, and indole-3-propionic acid (iPA) in different plants under drought stress [126].

Moreover, MT regulates the *HMA2*, *HMA3*, *HMA4*, *IRT1*, and *Nramp1* genes to mitigate the negative effects of Cd toxicity [43]. Adenosine triphosphate (ATP)-binding cassette (ABC) may facilitate the transfer of Cd stress to the xylem via the apoplast [130]; these genes may promote Cd sequestration in root vacuoles and stimulate its absorption and transfer to the xylem [130,131]. MT-upregulated and -downregulated genes are essential in cotton crops to mitigate the negative impacts of drought and Cd stress. However, the regulation of leaf senescence and ion transport in cotton under abiotic stress by these genes in response to MT application is still unclear.

## 5. Conclusions and Future Recommendations

The impact of abiotic stress on plant growth and yield severely threatens agricultural production, particularly cotton productivity. To alleviate the negative effects of abiotic stress on cotton, the plant uses a variety of biochemical, physiological, biochemical, and molecular responses [49].

Hormones such as MC and MT can decrease or alleviate the harmful effects of cotton under abiotic stress. The exogenous supply of both hormones is crucial for plant development and yield under abiotic stress conditions. Both hormones play indispensable roles in regulating crop metabolism and the complex processes of plant function; nevertheless, the roles of both hormones in the underlying processes of abiotic stress in cotton are still poorly understood.

It is confirmed that MC improves the functional phase of cotton leaves by increasing the CK contents during the middle and late stages of leaf development, decreasing GA production, and delaying the concentrations of ABA and ET. However, MC interference with CK, GA, ABA, and ET during the functional phase of leaves and its response to cold stress are attractive targets in cotton molecular research.

MC inhibits the activity of RuBisCo due to CO_2_ restriction, resulting in a decrease in photosynthesis. MC inhibits RuBisCo activity, limiting the photosynthetic activity and, as a consequence, yield. However, the effects of MC on this morpho-physiological activity in cotton during abiotic stress are unclear.

It has been confirmed that MC treatment increases endogenous ABA and auxin levels during the up-and downregulation of genes such as *GA2Ox, GA30x*, *GA2oox*, and *CPS* in root tissue. However, the underlying chemical process in cotton under abiotic stress remains unknown.

In response to drought stress, plants treated with MC may increase the expressions of genes involved in ABA synthesis and signaling. The increase in ABA content is related to an increase in stress resistance. However, the performance of MC on the underlying mechanism in cotton under various stresses is still unknown.

MT improves plant morpho-physiological activity via cell enlargement and osmotic stress reduction and by maintaining the water content in shoots, improving plant antioxidants, and regulating plant photosynthesis activity under Cd stress via the downregulation of Cd transporter genes, such as *LOC107894197*, *LOC107955631*, and *LOC107899273* in cotton.

It has been confirmed that MT decreases the negative effects of drought and Cd stress via the up- and downregulation of specific genes, such as *NCED1, NCED2, GhYUC5*, *GhGA3ox2*, *GhIPT2 HMA2*, *HMA3*, *HMA4*, *IRT1*, and *Nramp1*. However, the function of these genes in regulating leaf senescence and ion transport in cotton under other stresses is still unknown under MT application.

## Figures and Tables

**Figure 1 ijms-25-00235-f001:**
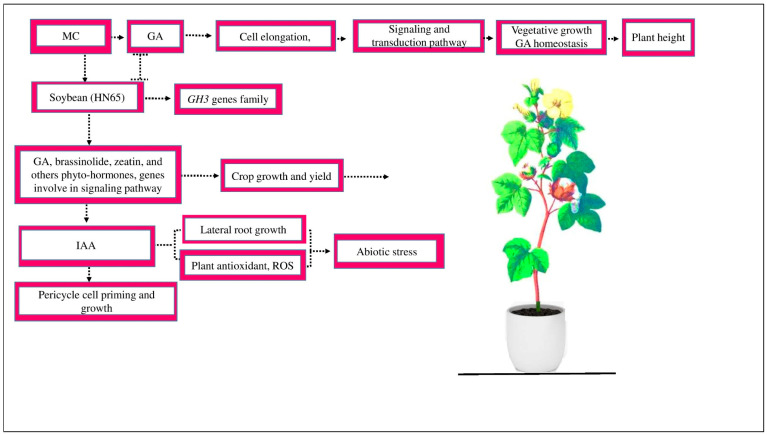
MC affects the GA activity involved in cell elongation and inhibits the signaling and transduction activation mechanism, which hinders vegetative development and GA homeostasis and, as a result, reduces plant height. MC affects the signaling of different hormones. In soybean variety HN65, during zeatin production, the expression of several genes, such as *GH3*, is downregulated. The genes responsible for GA and brassinosteroid biosynthesis are suppressed, which, as a result, control cotton growth and yield. MC affects the signaling of other hormones, such as IAA. IAA plays an essential role in pericycle cell priming and growth. IAA increases the number of lateral root growth, enhances plant antioxidants, and scavenges ROS under abiotic stress.

**Figure 2 ijms-25-00235-f002:**
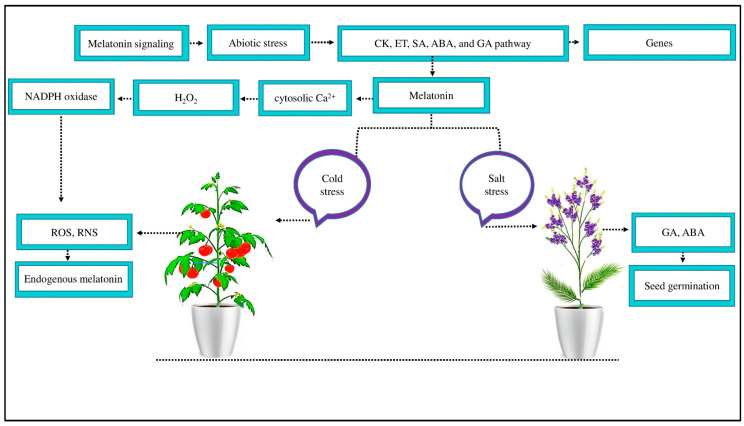
MT signaling mitigates the negative effects of abiotic stress. MT reduced abiotic stress by activating the signaling pathways of CK, ET, SA, ABA, and GA to induce several defense genes. An exogenous supply of MT reduces salt stress in *Limonium bicolor* by increasing GA and ABA concentrations and increasing seed germination. Likewise, MT reduced the negative effect of cold stress by scavenging ROS and RNS and improving endogenous MT. Moreover, MT acts as a signaling molecule at the plasma membrane, causing cytosolic Ca^2+^ to increase and signaling H_2_O_2_ accumulation through NADPH oxidase enzyme activity to scavenge ROS and RNS.

**Figure 3 ijms-25-00235-f003:**
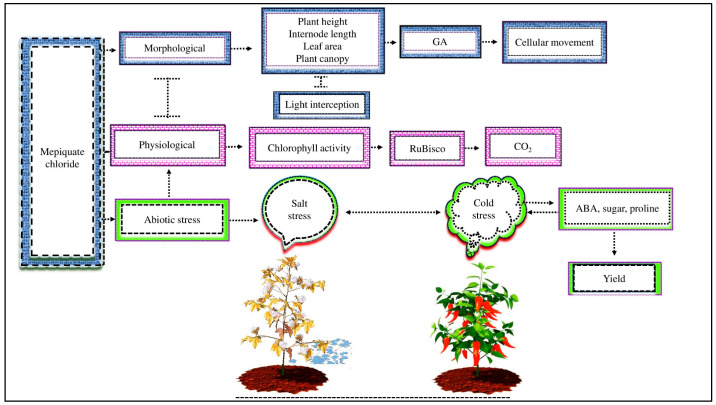
MC affects plant morpho-physiological activity under abiotic stress. MC application decreases plant height, internode length, leaf area, and plant canopy but increases light interception by inhibiting endogenous GA and disturbing cellular movements. MC lowers photosynthetic activity by decreasing the activity of RuBisco enzymes, which are involved in CO_2_ fixation during net photosynthesis. In addition, exogenous MC reduces salt stress in cotton and cold stress in sweet pepper by improving the ABA, sugar, and proline content and, as a result, improves plant yield.

**Figure 4 ijms-25-00235-f004:**
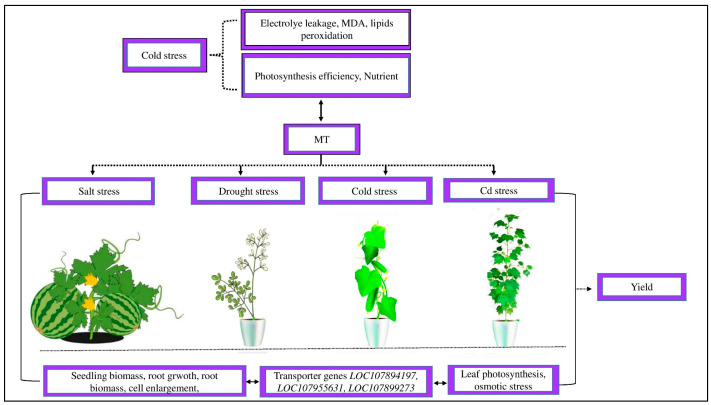
Abiotic stress affects plant morpho-physiological activity. Cold stress increased electrolyte leakage, MDA, and lipid peroxidation and reduced photosynthetic efficiency and the imbalance of nutrients. Exogenous MT decreased MDA formation, electrolyte leakage, and lipid peroxidation and increased the photosynthetic capacity and uptake of balanced nutrients. MT reduced salt stress in watermelons, drought stress in *Moringa oliguria*, cold stress in cucumbers, and Cd stress in cotton by improving the seedling biomass, root growth, root biomass, cell enlargement, leaf photosynthetic activity, osmotic stress, and transporter genes *LOC107894197*, *LOC107955631*, and *LOC107899273* and, as a result, improved the crop yield.

**Figure 5 ijms-25-00235-f005:**
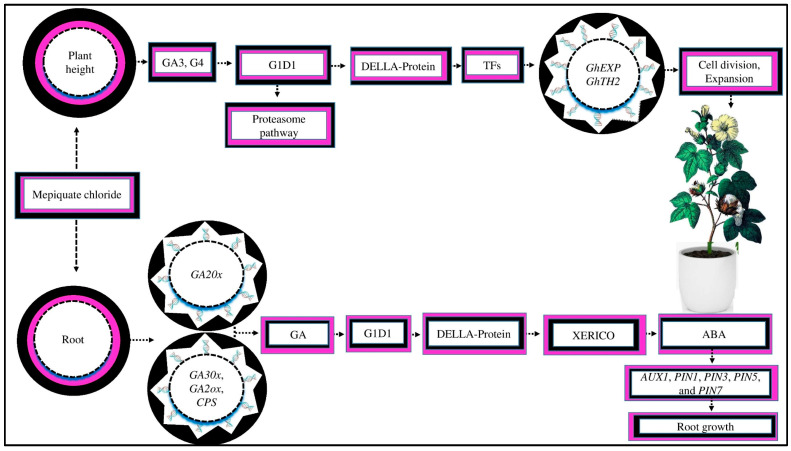
MC affects crop growth and yield by regulating gene expression. MC reduces the plant height by inhibiting GA_3_ and GA_4_. GAs interact with the G1D1 (gibberellin insensitive dwarf1) receptor via the proteasome pathway. GAs stimulate plant growth by increasing the degradation of DELLA protein (the key repressor of GA signaling). When the active GA content is reduced, DELLA proteins interact with transcription factors (TFs) to inhibit the upregulation of target genes, which suppresses cell division and expansion in cotton plants. Moreover, MC induces lateral root development by regulating endogenous GA. MC upregulates GA catabolism genes, such as *GA20x*, and downregulates expression genes, such as *GA30x*, *GA2oox*, and *CPS* in GA biosynthesis. The GA receptor GA1D1 causes DELLA protein degradation, which controls the production of the *XERICO* gene (a stimulator of ABA biosynthesis). MC decreases the endogenous level of ABA. The reduction in ABA promotes root development by upregulating auxin transporter genes, such as *PIN1*, *PIN3*, *PIN5*, *PIN7*, and *AUX1*.

**Figure 6 ijms-25-00235-f006:**
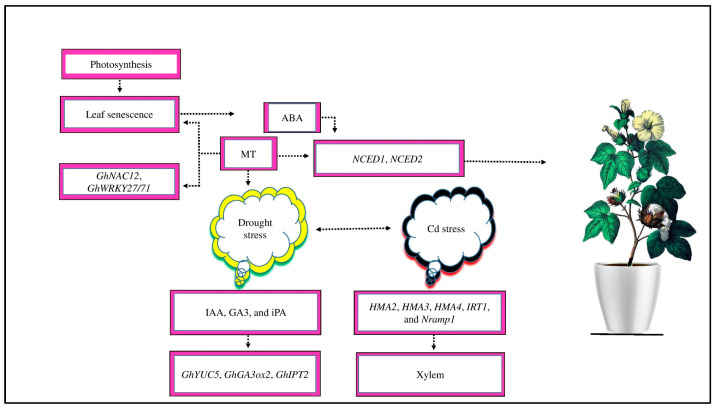
Cotton photosynthetic activity decreases as a result of leaf senescence. Exogenous MT prevents leaf senescence by the downregulation of chlorophyll degradation related-genes, such as *GhNAC12* and *GhWRKY27/71*. ABA is a priming hormone that promotes leaf senescence in response to abiotic stress. The contents of ABA and its biosynthesis genes, such as *NCED1* and *NCED2*, increase during leaf senescence in cotton. The supply of MT reduces the expression of these genes and their accumulation in cotton. Moreover, under drought stress, MT application delays leaf senescence by upregulating IAA, GA3, and CTK biosynthesis-related genes, such as *GhYUC5*, *GhIPT2*, and *GhGA3ox2*, which increases the endogenous levels of iPA, GA_3_, and IAA. Moreover, MT regulates *HMA2*, *HMA3*, *HMA4*, *IRT1*, and *Nramp1* genes to transport Cd stress from root vacuoles to xylem to mitigate the adverse effects of cadmium toxicity.

**Table 1 ijms-25-00235-t001:** MC and MT mitigate the negative effects of abiotic stress in cotton.

Hormones	Dose of MC and MT	Crops	Abiotic Stress	References
MC	50–150 g ha^−1^	Cotton	Enhanced salt tolerance	[37]
			Enhanced cold tolerance	[38]
			Heat stress	[39]
			Drought stress	[39]
			Enhanced drought tolerance	[40]
MT	50 μM to 100 μM	Cotton	Enhanced cold tolerance	[28,41]
			Enhanced Cd tolerance	[42]
			Enhanced Cd tolerance	[43]
			Enhanced cold tolerance	[44]
			Enhanced drought tolerance	[44,45]
			Enhanced salt tolerance	[41,46,47]
			Enhanced salt tolerance	[30,48]
			Ultraviolet stress	[49]
			Heavy metal stress	[49,50]

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
