# Peer review of "The Roles of Mepiquate Chloride and Melatonin in the Morpho-Physiological Activity of Cotton under Abiotic Stress"

_ijms, 2023, doi:10.3390/ijms25010235_

Round 1

Reviewer 1 Report

Comments and Suggestions for Authors

Dear Authors, 

My comments about your article are stated below. 

Best regards,

1. The title of the review is so long. For example, it could be “The role of mepiquate chloride and melatonin on morpho-physiological activity of cotton under abiotic stress"

2.The authors should add the abstract part as general information about the Mepiquate chloride and melatonin. Also add some information about cotton?

3. I recommend that in the review article, it should be stated which search engines and which years the literature review covers. The method should be explained.

3. Table 1 showed the MC and MT mitigates the negative effect of abiotic stress in different crops. I think that this table can be deleted because the review was written on cotton.

5. I can recommend that general information about other crops can be given in new subtitle instead of adding Tables 1 and 2. Otherwise, it is seen like comparing cotton with other plants.

6. Why did you write the sub title 2.2 MC induced genes involved in plant growth?  "Plant growth" should be changed as "cotton growth".

7. Why was the "Nevertheless, the underlying molecular process of this phenomenon in cotton under abiotic stress is still unknown" written at the end of some subtitles? This topic was discussed under "MT relevant genes involve in plant growth".

8. Subtitle 2.4 should be changed as "MT and MC relevant genes involve in cotton growth".

9. The authors should use "review" word instead of "study" word. Because this is a review.

10. Why did the authors write as following sentence or similar sentences during the review "However, the function of these genes 509 in regulating leaf senescence and ion transport in cotton under other stress is unknown 510 under MT application"? It can be written as 1 or 2 sentences in a part of the review.

11. I can recommend that part 2 should be changed with part 3. Because general information on MT and MC were given in part 3.

12. Generally, previous information should be briefly discussed under each section. Recommendations can be given in the conclusion and future recommendation part of the review.

Author Response

Response to reviewer 1

Dear Authors,

My comments about your article are stated below.

Ans. Respected reviewer, Okay, thank you very much for improving our work.

Best regards,

  1. The title of the review is so long. For example, it could be “The role of mepiquate chloride and melatonin on morpho-physiological activity of cotton under abiotic stress"

Ans. Dear reviewer thank you very much yes you are right. The title has been changed accordingly.

  1. The authors should add the abstract part as general information about the Mepiquate chloride and melatonin. Also add some information about cotton?

Ans. Dear reviewer, information about mepiquate chloride and melatonin has been added to the abstract section in lines 27 and 28, and in addition, information about cotton in line 12 has been added accordingly.

  1. I recommend that in the review article, it should be stated which search engines and which years the literature review covers. The method should be explained.

Ans. Dear reviewer thanks for your keen observation. We have used web of science, science direct and google scholar search engine and reviewed the published data of last decade for comprehensive overview to make this review article an innovative approach for readers.

  1. Table 1 showed the MC and MT mitigates the negative effect of abiotic stress in different crops. I think that this table can be deleted because the review was written on cotton.

Ans. Dear reviewer, Table 1 has been changed to the effect of MC and MT on cotton under abiotic stress in line 114. It’s perfect now. 

  1. I can recommend that general information about other crops can be given in new subtitle instead of adding Tables 1 and 2. Otherwise, it is seen like comparing cotton with other plants.

Ans. Dear reviewer, yes you are right. Table 1 has been changed, and table 2 has been deleted.

  1. Why did you write the sub title 2.2 MC induced genes involved in plant growth? "Plant growth" should be changed as "cotton growth".

Ans. Dear reviewer, sorry for the inconvenience. Subtitle 2.2 has been changed to MC-relevant genes involved in cotton growth accordingly.  The plant growth has been changed to cotton growth in line 384 and 385.

  1. Why was the "Nevertheless, the underlying molecular process of this phenomenon in cotton under abiotic stress is still unknown" written at the end of some subtitles? This topic was discussed under "MT relevant genes involve in plant growth".

Ans. Dear reviewer you are right. Actually, the underlying process of this phenomenon in cotton is studied in some stresses such as drought, salt high temperature, but some other stresses such as heavy metal, cold stress, nutrient stress etc (abiotic stress) are still less documented that’s why we used it. . .but according your comments I changed the sentence to “Nevertheless, the underlying molecular process of this phenomenon in cotton under other abiotic stresses is still unknown” to make it more clear in line 413. 

  1. Subtitle 2.4 should be changed as "MT and MC relevant genes involve in cotton growth".

Ans. Dear reviewer thank you. Both titles have been changed in line 384 accordingly.

  1. The authors should use "review" word instead of "study" word. Because this is a review.

Ans. Okay yes you are right. The word study has changed to review in most of the places where it's needed, such as in lines 15, 20, and 29.   

  1. Why did the authors write as following sentence or similar sentences during the review "However, the function of these genes 509 in regulating leaf senescence and ion transport in cotton under other stress is unknown 510 under MT application"? It can be written as 1 or 2 sentences in a part of the review.

Ans. Dear reviewer, yes you are right sorry for inconvenience. The repeated sentences from line no 488 has been deleted accordingly.

  1. I can recommend that part 2 should be changed with part 3. Because general information on MT and MC were given in part 3.

Ans. Dear reviewer, yes I agreed with you. Its seems perfect, thank you. Part 2 has been changed with part 3 but without any highlighting of the color of the replacement because the highlighting of the paragraph makes the paper corrections unclear. 

  1. Generally, previous information should be briefly discussed under each section. Recommendations can be given in the conclusion and future recommendation part of the review.

Ans. Dear reviewer, the review contains a lot of information. Further investigation in this field may increase the paper's length and may decrease the reader's interest. In conclusion the black color represents the conclusion while highlighted with red shows the future recommendation which will still be required in future. Thank you. 

Dear reviewer, thank you once again. If you need any further modification, please let us know. Thank you once again for improving our works. All your comments were constructive and appreciable.

Reviewer 2 Report

Comments and Suggestions for Authors

Dear Authors,

This is a well-performed work on the potential applications of mepiquate chloride and melatonin  on the potential effects reducing the effects of abiotic stress. The graphics are very nicely prepared and summarize each issue described.

I have two suggestion that the authors should comment on:

The literature should be checked and corrected (insert appropriate numbers in the Review text and correct the numbering in the list of references):

Line 58 – Raza et al. 2023 - please change to no. 13 - it is entered in the literature list as  75

Line 141 – Tung et al. 2020, please change it to 52 - it is entered in the literature list as 36

Line 345- Rademacher 2000 please change to 110 in the literature list it is marked as 112.

Please attach an additional column to the Tables, in which please include the MC or MT dose used. 

Author Response

Response to reviewer 2

Dear Authors,

This is a well-performed work on the potential applications of mepiquate chloride and melatonin on the potential effects reducing the effects of abiotic stress. The graphics are very nicely prepared and summarize each issue described.

Ans. Dear reviewer, thank you very much for your time and improving our work.

I have two suggestion that the authors should comment on:

The literature should be checked and corrected (insert appropriate numbers in the Review text and correct the numbering in the list of references):

Ans. Dear reviewer, the reference and text have been checked and rectified accordingly.

Line 58 – Raza et al. 2023 - please change to no. 13 - it is entered in the literature list as  75.

Ans. The Reference has been changed to 13.

Line 141 – Tung et al. 2020, please change it to 52 - it is entered in the literature list as 36

Ans. Thank you. Due to the change to the changing of section placement in article The reference Tung et al. 2020 has changed, and now its Reference 55 in line no 283.

Line 345- Rademacher 2000 please change to 110 in the literature list it is marked as 112.

Ans. Due to the changing the format and sections in article Rademacher 2000 has changed and now is reference 56.

Please attach an additional column to the Tables, in which please include the MC or MT dose used. 

Ans. Dear reviewer, the table has been changed. And the suitable dosage information has been added to the table. 

Round 2

Reviewer 1 Report

Comments and Suggestions for Authors

Dear Authors,

The requested corrections have been made. The publication is acceptable in its present form.

Best regards,